# Validation of Reference Genes for Quantitative PCR in Johnsongrass (*Sorghum halepense* L.) under Glyphosate Stress

**DOI:** 10.3390/plants10081555

**Published:** 2021-07-28

**Authors:** María Noelia Ulrich, Esteban Muñiz-Padilla, Alejandra Corach, Esteban Hopp, Daniela Tosto

**Affiliations:** 1Instituto de Agrobiotecnología y Biología Molecular (IABIMO), Instituto Nacional de Tecnología Agropecuaria (INTA), Consejo Nacional de investigaciones Científicas y Tecnológicas (CONICET), Nicolás Repetto y de los Reseros S/N, Hurlingham B1686IGC, Argentina; corach.alejandra@inta.gob.ar; 2Facultad de Ciencias Agropecuarias, Universidad Nacional de Entre Ríos/Facultad de Ciencia y Tecnología, Universidad Autónoma de Entre Ríos, Ruta 11 km 10.5, Oro Verde ER E3100XAD, Argentina; esteban.muniz@fca.uner.edu.ar; 3Departmento de Fisiología y Biología Molecular y Celular, Facultad de Ciencias Exactas y Naturales, Universidad de Buenos Aires, Ciudad Universitaria, Buenos Aires C1428EGA, Argentina; hopp.esteban@inta.gob.ar

**Keywords:** weeds, herbicide resistance, RT-qPCR, gene expression

## Abstract

Weeds are one of the main causes of the decrease in crop yields, with Johnsongrass (*Sorghum halepense* L.) being one of the most significant. Weeds can be controlled by herbicides, but some have developed resistance. Quantitative PCR is the technique of choice for studying gene expression related to herbicide resistance because of its high sensitivity and specificity, although its quantitative accuracy is highly dependent on the stability of the reference genes. Thus, in this study we evaluated the stability of different reference genes of glyphosate-resistant *S. halepense*. Nine genes frequently used as reference genes were selected: MDH, ADP, PP2A, EIF4α, ACT, ARI8, DnaJ, Hsp70, and ALS1, and their expression analyzed in susceptible and resistant biotypes at 0, 24 and 72 h post-application of glyphosate. The stability was analyzed with the geNorm, NormFinder, and BestKeeper software programs and using the ΔCt method. RefFinder was used to generate a comprehensive stability ranking. The results showed that PP2A and ARI8 were the most stable genes under the test conditions. EPSPS expression was also verified against the best two and the worst two reference genes. This study provides useful information for gene expression analysis under glyphosate stress and will facilitate resistance mechanism studies in this weed species.

## 1. Introduction

Weeds are one of the main biotic threats to crops, capable of causing a loss of up to 34% of crop yields worldwide, due to the competition for inorganic nutrients, light, and water [1]. Although herbicides can control weeds efficiently and economically, enhance crop yield, and liberate the workforce, over-reliance on herbicides has resulted in the evolution of herbicide resistance in weed species [2].

To date, at least 262 weed species (152 dicots and 111 monocots) have become resistant to 167 different herbicides and to 23 of the 26 known herbicide sites of action, and herbicide-resistant weeds have been reported in 94 crops in 71 countries [3].

One of the most troublesome weeds worldwide is Johnsongrass (*Sorghum halepense* (L.) Pers), which belongs to the family Poaceae [4]. This weed is a C4 perennial and rhizomatous grass weed [5] that has spread from its hypothesized west Asian center of origin across much of Asia, Africa, Europe, North and South America, and Australia [6]. It was introduced as a forage grass into the USA early in the 19th century and into Argentina around 1910 [5]. However, it was declared an agricultural weed in 1930, and as a result its planting and multiplication were banned in Argentina in 1951 [7]. At present it is on the top-10 list of weeds that have grown the most in recent years and is more geographically dispersed [8].

The main strategies of reproduction of this weed in agricultural ecosystems are clonal dispersion of rhizomes and seed production. For decades this weed has been controlled by herbicides, mainly glyphosate [9]. Early studies and speculation suggested that the evolution of glyphosate resistance in weeds would be extremely rare [2]. However, those studies had been performed in insects and fungi, which are organisms that have much shorter life cycles than weeds. Conversely, the development of several transgenic glyphosate-resistant crops and the increase in glyphosate use in agriculture have provided insights into the many existing routes of resistance, including novel molecular genetic mechanisms [10]. In Argentina, the first case of *S. halepense* resistant to glyphosate was recorded in 2005 [11], and new cases of resistance have been reported in different ecoregions of the country since then. Reported cases of glyphosate-resistant *S. halepense* are variable worldwide. Biotypes resistant to different groups of herbicides, such as the Herbicide Resistance Action Committee (HRAC) group 9 (glyphosate), HRAC group 1 (e.g., haloxyfop-methyl, clethodim, propaquizafop, etc.), HRAC group 2 (e.g., nicosulfuron, foramsulfuron, imazethapyr, etc.), and HRAC group 3 (pendimethalin), have also been reported [3]. In Argentina, two multi-resistant *S. halepense* have been reported: one biotype resistant to two sites of action, glyphosate and haloxyfop-methyl [12], and a triple-resistant biotype, resistant to glyphosate, clethodim and haloxyfop-methyl [13]. In Europe, the first case of glyphosate-resistant *S. halepense* has been recently reported [14]. The appearance and dispersal of resistant weeds lead to an increase in the production costs of the affected lots, because the treatment to control resistant weeds signifies 31% of the total cost of protecting the crop [15].

Resistant weeds can survive herbicide applications by a variety of mechanisms, which can be divided into two broad categories: target-site resistance (TSR) and non-target-site resistance (NTSR) mechanisms. TSR mechanisms often involve mutations in genes encoding the protein targets of herbicides or increased amounts of protein target because of increased gene expression or gene duplication, whereas NTSR mechanisms include reduced absorption or translocation and increased sequestration or metabolic degradation of the herbicide [16,17].

The development of effective and sustainable weed management strategies thus requires knowledge of weed biology and ecology [18]. In addition, clarifying the resistance mechanism of weeds is essential for their control and to ensure crop yield and the sustainable application of herbicides [19]. The amplification of the target site and the over-expression of detoxification genes often involved in herbicide resistance can be evaluated by gene expression analysis [20,21]. The most efficient approach for quantification of gene expression is quantitative real time PCR (qPCR) because of its simplicity, sensitivity, accuracy, and cost; however, this method is often limited due to the absence of consistent reference genes for data normalization [22]. The most reliable method of normalization of samples is the parallel quantification of endogenous reference genes [23].

Normalization during qPCR analysis usually relies on using a reference gene expressed at stable levels, regardless of the experimental conditions, to ensure the veracity of the qPCR analysis [24]. The ideal reference gene should be expressed stably across all samples and groups. However, there is no universal gene that satisfies the requirements, and so the reference gene of choice becomes dependent on factors such as the sample and tissue type, the experimental conditions, and the integrity of the sample [25]. Several groups have searched for suitable reference genes for expression analysis under different conditions in plants, including weeds [26,27,28,29]. A number of recent studies have focused on the stress under herbicide treatment for weeds [21,30,31,32,33].

Researchers have developed a series of algorithms to facilitate the analysis of multiple reference genes and allow a comparison of the stability of these genes. Three of these algorithms, BestKeeper, geNorm, and NormFinder, are combined in the free web tool RefFinder [34], and, in combination with a fourth comparison (ΔCt method [35]), enable assessment of the most stable reference gene. These tools have been widely used to validate reference genes in different plants, such as *Eleusine indica*, *Conyza bonariensis*, *Conyza canadensis*, *Alopecurus myosuroides*, *Sorghum bicolor*, *Avena fatua*, *Salix matsudana*, and *Lilium* spp., under different experimental conditions, including drought stress, salt stress, heavy metal stress, and herbicide stress [21,22,27,30,31,32,33,36].

Based on the above, the aim of this study was to assess nine common reference genes (MDH, ADP, PP2A, EIF4α, ACT, ARI8, DnaJ, Hsp70, and ALS1) in search of validation of stably expressed reference genes in *S. halepense* under glyphosate stress.

## 2. Results

### 2.1. Primer Specificity and PCR Amplification Efficiency

The specificity of the primer pairs was adequate, as confirmed by agarose gel electrophoresis (Figure 1) and by the presence of a single peak in melting curves (Figure 2). The primers generated amplicons of the reference genes that ranged from 103 to 150 bp. These amplicons were sequenced and compared with GenBank sequences using the BLASTN algorithms to verify the amplicon specificity of the targeted gene in *S. halepense*. The amplification efficiency (Eff%) ranged from 97% (ACT) to 106.8% (ALS1) and the regression coefficient of the standard curve varied from 0.934 (ALS1) to 1 (DnaJ) (Table 1).

The quantification cycle (Cq) values of the nine candidate genes across all samples ranged from 19.4 to 29.7 (Figure 3). The mean Ct values of the genes evaluated ranged from 20.90 (MDH) to 26.63 (Hsp70). The low Ct values of MDH corresponded to a high level of expression, whereas the higher Ct values of Hsp70 corresponded to the lowest expression. The Ct values for ALS1 (22.22–29.35) and ACT (22.08–28.57) showed the largest variation for one gene, whereas those for DnaJ (20.07–22.41) and ARI8 (22.77–25.99) showed the smallest variation.

### 2.2. Analysis of Gene Expression Stability Using Different Software Programs

#### 2.2.1. BestKeeper

The type of analysis performed by BestKeeper differs from that performed by geNorm and NormFinder, because BestKeeper is based on the standard deviation (SD) and coefficient of variation (CV). The lowest SD and CV values indicate the most stable reference gene. Furthermore, only reference genes with SD values below 1 are acceptable. Under the conditions of this study, ACT and ALS1 yielded SD values above the acceptable values and, therefore, were considered not suitable; the genes with the lowest SD values were DnaJ, ARI8, and PP2A (0.43, 0.55, and 0.6, respectively) (Table 2).

#### 2.2.2. geNorm

The geNorm software identifies the optimal reference gene pair; smaller M values correlate with more stable gene expression. The reference gene was accepted only with M values below 1.5. Under glyphosate stress, PP2A and MDH ranked as the most stable pair of reference genes (M = 0.45), whereas ACT was the least stable gene (M = 1.27) and DnaJ did not meet the acceptable values because its M value was greater than the cut off (M > 1.5) (Table 2 and Figure 4a).

#### 2.2.3. NormFinder

The NormFinder software evaluates the most stable candidate gene by means of the stability value. The results include the most stable gene and the best combination of two reference genes. In this study, at the three times evaluated (i.e., 0, 24 and 72 h post-application of glyphosate), the most stable gene was PP2A (0.26) and the best combination of two genes was PP2A and ARI8 (Table 2 and Figure 4c), whereas the least suitable genes were ALS1 and ACT.

#### 2.2.4. ΔCt Method

The comparative ΔCt method identifies potential reference genes by comparing the relative expression of gene pairs within each sample [35] (Table 2 and Figure 4b). This analysis indicated that ARI8 and PP2A were the most stable genes in this study.

#### 2.2.5. RefFinder Tool

The gene ranking orders generated by geNorm, NormFinder, BestKeeper and the comparative ΔCt method showed some differences. To provide a comprehensive evaluation of the candidate reference genes, we performed another analysis using the web-based tool RefFinder, which integrates the algorithms. In this analysis, PP2A and ARI8 ranked as the most stable genes, whereas ACT and ALS1 were the least stable ones (Table 3).

### 2.3. Expression Level of EPSPS

The validation of the candidate reference genes was evaluated by analyzing the expression of EPSPS, which is the target enzyme of glyphosate, under the different conditions at 24 and 72 h post-application of glyphosate. The results showed that EPSPS expression under glyphosate stress did not significantly differ from the normalization with PP2A or ARI8. By contrast, EPSPS expression pattern was quite different when ACT and ALS1, the least stable genes, were used; indeed, in this case the outcome was a 2- to 4-fold downregulation of EPSPS (Figure 5).

## 3. Discussion

*Sorghum halepense* is one of the 10 worst weeds worldwide. One common measurement of control is the use of herbicides such as glyphosate, an herbicide of high impact due to the development of genetically modified organisms. However, to date, 43 weed species, including *S. halepense*, have developed resistance to glyphosate [3]. Thus, the appearance of glyphosate-resistant populations has made its control difficult. Altered transcriptional responses to treatments can provide important clues to the mechanism underlying the biological response of herbicide tolerance. The assessment of the levels of expression of genes related to herbicide resistance in weeds is essential, and the qPCR method is the test of choice for this purpose [21,30,31]. To determine the molecular basis of resistance, through investigation of the expression levels of either herbicide target genes or genes involved in detoxification, researchers need to first identify appropriate reference genes under the stress conditions of interest. Comparison and selection of these genes is a fundamental step in detecting variation in target genes [37]. In addition, the amplification efficiencies of the target and reference genes must be approximately equal to ensure an accurate comparison [24,38].

In this study, we first searched for putative reference genes for *S. halepense* and selected nine common reference genes: MDH, ADP, PP2A, EIF4α, ACT, ARI8, DnaJ, Hsp70 and ALS1. We then designed some primers and verified that they all satisfied the necessary criteria of efficiency for real-time qPCR. The reliability of the candidate reference genes was tested by three different software packages (BestKeeper, geNorm and NormFinder) and a web tool.

Analysis with BestKeeper showed that ACT and ALS1 should be excluded due to their SD values (SD > 1). For the rest of the algorithms, these two genes, which are often used as reference genes, were also in the worst positions in terms of gene expression stability, despite the fact that, in some validation assays, these genes have been found to be the most stable, e.g., in *Eleusine indica* under herbicide stress; in *Salix matsudana* under abiotic stress; and in *Conyza canadiensis* and *Conyza bonariensis* under glyphosate stress [21,32,33]. Other studies have demonstrated that the traditional reference gene ACT, which is involved in cytoskeleton structure, was changed by the experimental treatment, e.g., in *Galium aparine* [39] and *Avena fatua* [31].

With the sample set analyzed here, two of the algorithms identified PP2A and ARI8 as the most stable reference genes. This finding is consistent with previous studies on *Sorghum bicolor*, in which PP2A was the reference gene that responded more stably under different abiotic stresses [22]. In *Brassica napus*, PP2A also ranked among the top four reference genes studied [40].

All these results suggest that evaluating reference gene stability is indispensable prior to the analysis of target gene expression under specific experimental conditions [33], since the commonly used reference gene does not necessarily work appropriately in all species.

It is important to note that the results obtained differed according to the algorithms used for the assessment of stability. This variation is due to their different mathematical foundations [27]. We therefore used the RefFinder platform, a popular free tool for reference gene validation, which performs a quick analysis using the three algorithms for the validation of reference genes, by starting from a single input of the Cq values only [23].

The validation of the candidate reference genes was also evaluated by analyzing the expression of EPSPS, the molecular target of glyphosate in the shikimate pathway [9]. Some glyphosate-resistant weeds, such as *Lolium perenne* [41], *Eleusine indica* [21] and *Conyza canadensis* [20], overexpress EPSPS. In our study, the expression of EPSPS showed no increase after treatment with glyphosate, according to analysis of the best reference genes (ARI8 and PP2A); likewise, in *Coniza sumatrensis* [42] and *Amaranthus tuberculatus* [43], other glyphosate-resistant weeds, EPSPS expression shows no variation. The use of ACT or ALS1 as reference genes revealed an under-expression of our gene of interest. This finding indicates that it is imperative to have an appropriate reference gene for the accuracy of the results.

In summary, in the present study, we analyzed the transcription level of nine genes in *S. halepense* under glyphosate stress conditions. This study represents the first attempt to select a set of candidate reference genes in *S. halepense* under glyphosate stress for the normalization of gene expression data using qPCR; ARI 8 and PP2A were the reference genes with the best performance. The results of this study will facilitate future studies to understand the molecular mechanisms of glyphosate resistance.

## 4. Materials and Methods

### 4.1. Plant Material

*Sorghum halepense* resistant and susceptible genotypes were collected from the field. The original location of the rhizomes was recorded through GPS (Table 4). Rhizomes were grown in a glasshouse for two months during the autumn, under a 16 h/8 h photoperiod, at 24 °C ± 3 °C and 140 μmol/ m^−2^ s^−1^ light intensity.

Genetically identical plants at the same growth stage (3–4 leaves) were sprayed with glyphosate (Glyphosate SL at 48% w/V) at rates of 360 g ae ha^−1^. The samples were collected at three different times: before application (0 h post-application) and at 24 and 72 h post-application.

Leaf tissues were obtained from six plants: two sensitive (S1 and S2) and four resistant plants (R1, R2, R3 and R4). For each of these plants, three clones were made by vegetative propagation of the rhizomes (biological replicates). The samples collected were immediately frozen in liquid nitrogen and stored at −80 °C until RNA extraction.

Clones from sensitive plants were used as controls to evaluate herbicide application efficacy. All clones were grown, sprayed, and collected at the same time and under the same conditions.

### 4.2. RNA Isolation and cDNA Synthesis

Total RNA was extracted from leaves using the TRIzol protocol (Invitrogen, Carlsbad, CA, USA). RNA integrity was confirmed by 1.0% agarose gel electrophoresis, whereas RNA quality and quantity were determined using a NanoDrop™ 1000 Spectrophotometer (Thermo Fisher Scientific Inc., Waltham, MA, USA). Only the RNA samples with A260/A280 values ranging from 1.80 to 2 and A260/A230 values above 2 were used for cDNA synthesis. Genomic DNA was eliminated using DNase I (Invitrogen) for 20 min at room temperature. For each sample, 1 μg of total RNA was reverse-transcribed using a SuperScript IV transcription kit (Invitrogen, Waltham, MA, USA) and random hexamer primers according to the manufacturer´s instructions. The samples were stored at −80 °C until use.

### 4.3. Reference Gene Selection and Primer Design

Nine candidate reference genes selected for the study were identified from the literature: MDH, ADP, PP2A, EIF4α, ACT, ARI8, DnaJ, Hsp70, and ALS1 (Table 1). The primers of MDH, ADP, PP2A and EIF4α were obtained from a previous study [15], whereas the remaining sequences were obtained from *Sorghum bicolor* v3.1.1 using the platform (https://phytozome.jgi.doe.gov, accessed on 13 March 2019). The primers for qPCR were designed using the Primer3 online software (http://primer3.ut.ee/, accessed on 30 March 2019), with the following parameters: 55–65 °C annealing temperature, 18–25 bp primer length, 45–55 GC content, three maximum GC at the 3′ end, and 90–150 bp amplicon length; other parameters were used by default by the software. The primers were designed based on the CDS sequence, and it was verified that the hybridization of the forward and reverse primers was within the same exon. Similar analysis was undertaken for MDH, ADP, PP2A and EIF4α [22], by aligning the sequence in which the primers were reported (i.e., XM_002441244, XM_002453490, XM_021459375 and XM_002467034, Table 1) with the sequence obtained by homology from *Sorghum bicolor* v3.1.1 genome. In this case, ADP, PP2A and EIF4 primers hybridized within a single exon whereas for MDH, the forward and revere primers belong to different exons. The specificity was then checked in silico, using Primer BLAST. All primers have been aligned on target sequences. Primer specificity was assessed by performing common PCR using cDNA of *Sorghum halepense* as template and running the amplified product on a 2% agarose gel (Figure 1). Amplicons were sequenced to confirm the amplification of the targeted gene in *S. halepense*.

### 4.4. qPCR Assay

The shape of the dissociation curve was analyzed to confirm that each product had a single peak melting curve (Figure 2).

The amplification efficiency of each candidate gene was assessed by pooling the same volume of cDNA samples. The pool was diluted and used to generate four-point standard curves based on a ten-fold dilution series (1; 1/10; 1/100 and 1/1000). A non-template control was included in each amplification to monitor contamination and specificity.

The correlation coefficients (R^2^) and slope values were obtained from the standard curves, and the PCR amplification efficiencies (E) were calculated according to E = 10 ^−1/slope^.

The qPCR reactions were performed in a 20 μL total volume containing 10 ng cDNA, ROX 1X (Invitrogen, Waltham, MA, USA), SYBR Green 2X (Invitrogen, Waltham, MA, USA), 0.2 μM of each primer, 2 mmol/L MgCl2, 0.2 mmol/L of each dNTP, 1X Reaction buffer, Taq platinum 1 U (Invitrogen, Waltham, MA, USA) and nuclease-free water, using StepOne Plus (Applied Biosystems, Waltham, MA, USA), with the following conditions: 10 min at 95 °C, 40 cycles of 95 °C for 15 s, and 60 °C for 1 min. Amplicon specificity was verified by melting curve analysis (60 °C to 95 °C). The qPCR assay was carried out using three biological replicates for each condition and two technical replicates.

### 4.5. Data Analysis for Expression Stability

The box plot was plotted from the Ct of the biological replicates for each of the plants by averaging the technical replicates against each other.

The software programs geNorm [44], NormFinder [45] and BestKeeper [46] and the comparative ΔCt method [35] were selected to calculate the expression stability of the nine candidate reference genes. The web tool RefFinder [34] was then selected to generate a comprehensive ranking for the reference genes through comparison of the results calculated by these three software packages.

The BestKeeper algorithm depends on the SD and CV. The lowest SD and CV indicate the most stable reference gene. Only genes with SD < 1 were considered suitable as reference genes [46].

The Cq values of all reference genes used in geNorm and NormFinder were converted into relative quantities according to the formula 2^−ΔCt^ (ΔCt = the corresponding Cq value−minimum Cq). The mean values for the biological replicates were used as the input data for the geNorm and NormFinder analysis.

The NormFinder software evaluates the most stable candidate gene by means of the stability value, with the lower stability value indicating higher expression stability [38]. In comparison, the geNorm software evaluates the stability level by parameter M, where the lowest M value refers to the most stable gene expression [44].

The comparative ΔCt method examined the relative expression between gene pairs [35].

### 4.6. Validation of Reference Genes

The reliability of the reference genes obtained was validated by normalizing the expression of the target gene EPSPS (5-enolpyruvylshikimate-3-phosphate synthase), under the different experimental conditions, and using the combination of the two best reference genes and the most variable genes. LinRegPCR [47] was used to analyze the data obtained. The relative number of copies of the EPSPS gene was determined according to R = Eff _EPSPS_ ^(Ct C−Ct T)^/Eff _REF_ ^(Ct C−Ct T)^ [39] and analyzed using fgStatistics [48].

## Figures and Tables

**Figure 1 plants-10-01555-f001:**
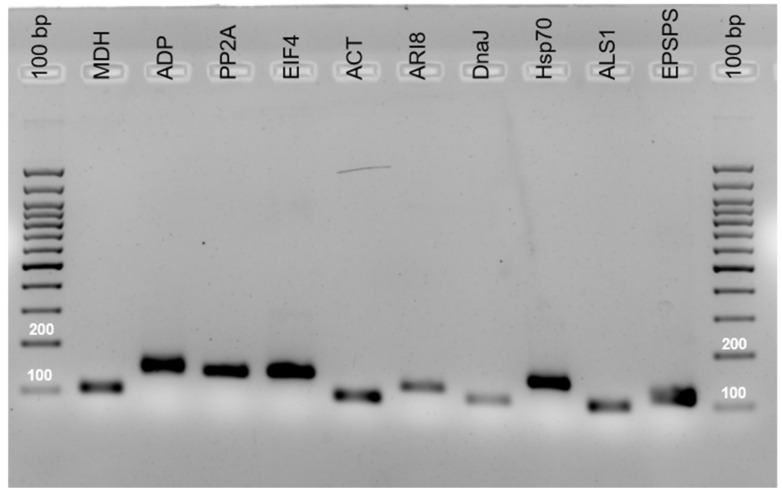
Amplification products of the nine candidate reference genes.

**Figure 2 plants-10-01555-f002:**
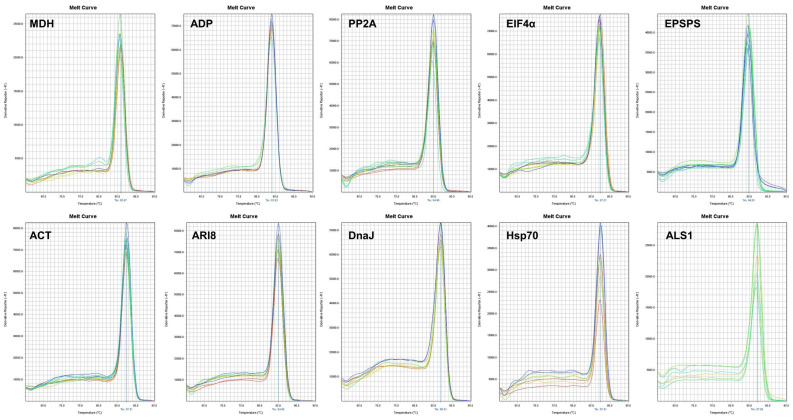
Dissociation curves of the nine reference genes evaluated and the EPSPS gene under experimental conditions, each showing a single peak.

**Figure 3 plants-10-01555-f003:**
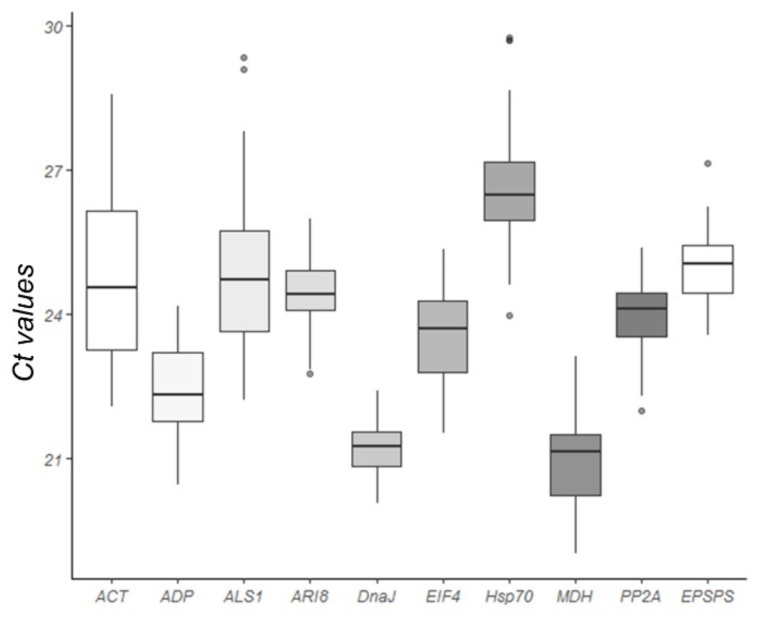
Expression levels of the nine candidate reference genes evaluated under glyphosate stress. Cycle threshold (Ct) values of the nine candidate reference genes in all samples. A line across the box value displays the median value of the Ct in the Box-plot graph. The lower and upper boxes indicate the 25th percentile to the 75th percentile, whereas the whiskers indicate the ranges for all samples.

**Figure 4 plants-10-01555-f004:**
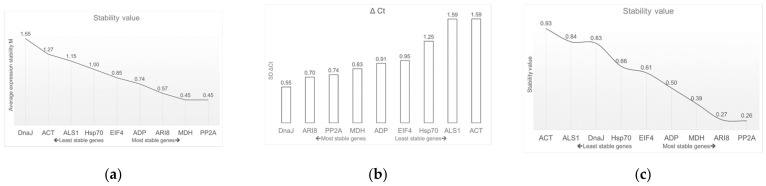
Values of gene expression stability of the nine candidate genes. (**a**) geNorm expression and stability ranking. (**b**) ΔCt method values and (**c**) NormFinder expression and stability ranking.

**Figure 5 plants-10-01555-f005:**
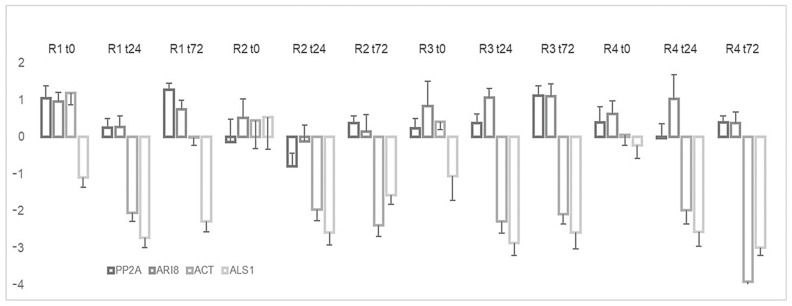
Relative expression of EPSPS under glyphosate stress and normalization to different reference genes.

**Table 1 plants-10-01555-t001:** Information on the primer pairs selected and amplification details. Slope: slope of standard curve; R^2^: correlation coefficient value for standard curve; Eff%: PCR efficiency [%]; E: primer efficiency calculated according to E = 10^−1/slope^.

Gene Symbol	Cellular Function	Accession Number	Primers (F/R) (5″–3″)	Amplicon Length (bp)	Standard Curve Parameters			
					Slope	R^2^	Eff%	E
MDH	Malate dehydrogenase	XM_002467034	TGCAGTGGTGGTGAATGGAA	103	−3.277	0.985	104.133	2.019
			GCGTCTTCTCTTCCGACAGC					
ADP	ADP-Ribosylation Factor	XM_002441244	GTCTGTCGGATGTGGGGATGT	136	−3.287	0.996	101.179	2.015
			CACAGCACACAGTCGGACATG					
PP2A	Serine/threonine Protein Phosphatase	XM_002453490	AACCCGCAAAACCCCAGACTA	138	−3.188	0.998	105.919	2.059
			TACAGGTCGGGCTCATGGAAC					
EIF4α	Eukaryotic Initiation Factor 4A	XM_002451491	CAACTTTGTCACCCGCGATGA	144	−3.316	0.995	100.265	2.002
			TCCAGAAACCTTAGCAGCCCA					
ACT	Actin	Sobic.009G005900.1	TCGAGACACTTGTGGCAGATT	100	−3.396	0.997	97.003	1.970
			CGCACATGGAGCCACAACAT					
ARI8	E3 ubiquitin protein ligase ARI8	Sobic.006G131000.2	CGGGCTCTGGAAACTGGATT	121	−3.327	0.999	99.772	1.998
			TTGATGCCCTGTTCTTGCCA					
DnaJ	Chaperone protein DnaJ 49	Sobic.003G185200.1	TTTCAGGACTGGTGGGATGC	103	−3.363	1	98.3	1.983
			GAGCAACAGCAGCAGTAGGA					
Hsp70	Heat shock 70 kDa protein	Sobic.002G249800.1	ACCTGCTGAAGTCACCAAGG	150	−3.178	0.996	106.392	2.064
			CCACCACCTTGTTGCATGTG					
ALS1	Acetolactate Synthase	Sobic.004G155800.2	TGGGCCTTGGCAATTTCC	100	−3.168	0.934	106.827	2.089
			AGATCCGCCTTATCCACTGCAT					
EPSPS	5-enolpyruvylshikimate-3-phosphate synthase	Sobic.010G023800.1	CATGGACCGAGACTAGCGTAACTG	113	−3.309	0.979	100.538	2.005
			AGTCATGGCAACATCAGGCATT					

**Table 2 plants-10-01555-t002:** Gene expression stability ranked by BestKeeper, geNorm, NormFinder, ΔCt method and RefFinder. SD (±Cq): standard deviation of the Cq; CV (% Cq): coefficient of variation expressed as percentage of the Cq level.

Rank	BestKeeper			geNorm		NormFinder		ΔCt Method		RefFinder	
	Gene	SD (±Cq)	CV (% Cq)	Gene	Stability	Gene	Stability	Gene	Stability	Gene	Stability
1	DnaJ	0.43	2.04	PP2A	0.45	PP2A	0.26	DnaJ	0.55	PP2A	1.32
2	ARI8	0.55	2.26	MDH	0.45	ARI8	0.27	ARI8	0.70	ARI8	2.21
3	PP2A	0.60	2.53	ARI8	0.57	MDH	0.39	PP2A	0.74	DnaJ	2.63
4	MDH	0.69	3.28	ADP	0.74	ADP	0.50	MDH	0.83	MDH	2.63
5	ADP	0.75	3.34	EIF4	0.85	EIF4	0.61	ADP	0.91	ADP	5.00
6	EIF4	0.82	3.47	Hsp70	1.00	Hsp70	0.66	EIF4	0.95	EIF4	6.00
7	Hsp70	0.93	3.50	ALS1	1.15	DnaJ	0.83	Hsp70	1.25	Hsp70	7.00
8	ALS1	1.23	4.96	ACT	1.27	ALS1	0.84	ALS1	1.59	ALS1	8.00
9	ACT	1.36	5.49	DnaJ	1.55	ACT	0.93	ACT	1.59	ACT	9.00

**Table 3 plants-10-01555-t003:** Expression stability ranking of the nine candidate reference genes as calculated by RefFinder.

Rank	BestKeeper		geNorm		NormFinder		ΔCt Method		RefFinder	
	Gene	SD (±Cq)	Gene	Stability	Gene	Stability	Gene	Stability	Gene	Stability
1	DnaJ	0.36	PP2A	0.49	PP2A	0.22	PP2A	0.94	PP2A	1.32
2	ARI8	0.48	MDH	0.49	ARI8	0.35	ARI8	1.00	ARI8	2.21
3	PP2A	0.56	ARI8	0.59	MDH	0.50	DnaJ	1.01	DnaJ	2.63
4	MDH	0.64	DnaJ	0.64	DnaJ	0.51	MDH	1.04	MDH	2.63
5	ADP	0.66	ADP	0.73	ADP	0.82	ADP	1.17	ADP	5.00
6	EIF4	0.69	EIF4	0.80	EIF4	0.95	EIF4	1.23	EIF4	6.00
7	Hsp70	0.89	Hsp70	0.95	Hsp70	1.28	Hsp70	1.51	Hsp70	7.00
8	ALS1	1.16	ALS1	1.11	ALS1	1.31	ALS1	1.54	ALS1	8.00
9	ACT	1.35	ACT	1.2	ACT	1.46	ACT	1.64	ACT	9.00

**Table 4 plants-10-01555-t004:** Genotype sampling sites.

ID	Collection Site	Latitude	Longitude
S1	Oro Verde, Entre Ríos	31°49′58″ S	60°31′27″W
S2	Facultad; Entre Ríos	31°49′59″ S	60°31′ 28″ W
R1	Soresi; Entre Ríos	31°19′56″ S	60°01′16″ W
R2	Pavioti; Entre Ríos	31°18′48″ S	59°46′30″ W
R3	Hasenkamp; Entre Ríos	31°27′20″ S	59°52′54″ W
R4	Hernandarias; Entre Ríos	31°17′11″ S	59°46′42″ W

## Data Availability

Not applicable.

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
