# Peer review of "Validation of Reference Genes for Quantitative PCR in Johnsongrass (Sorghum halepense L.) under Glyphosate Stress"

_plants, 2021, doi:10.3390/plants10081555_

Round 1

Reviewer 1 Report

Dear Authors,

I focus on materials and methods:

  1. missing data on growing conditions of plants
  2. Line 241 innapropiate English "Genes with stable expression..."
  3. Time points of sampling
  4. Sampling before treatment?

Reviewer 2 Report

Review of the manuscript
Validation of reference genes for quantitative PCR in John-songrass (Sorghum halepense L.) under glyphosate stress
The presented work fits perfectly into the general trend to understand the cause, nature, genetics, mechanism, and solutions of herbicide-resistant weeds. In this context, it is worthy of publication. Furthermore, the paper is well written and conforms to the standards of selecting samples and statistical analyses accepted for papers aiming to develop reference genes for qPCR. Below I have some comments that authors might consider whether to incorporate into their text.
Abstract
The abstract includes all required sections. The authors should mention in the abstract that they performed the analyses on a particular developmental stage and vegetative tissue of Sorghum.
Introduction
l. 42-44, l. 176 research object - The authors focused their paper on the plant Sorghum halepense L. the plant often used for erosion control but now common in many ecosystems, including crop fields and pastures in almost every corner of the world. In this context, the authors should present their research object in more detail (e.g., How much does its presence impact agronomy? Where it comes from (native range)? To which group of herbicides it shows resistance?
Material and Methods
The authors well-described quality assessment of RNA templates, reverse transcription qPCR conditions, PCR amplification efficiency, and data analysis, including confidence estimation and software used. However, the authors should additionally provide the following information:
l. 238 – where they collected the plants in the field (resistant and nonresistant).
l. 239 – what were the growing conditions in the greenhouse (temperature, light exposition, etc.)? How long did plants grow under those conditions (season, time)?
l. 240 and l. 244 – unclear terminology used - if the authors studied three independent biological replicates, then these samples could not be genetically identical. Perhaps rearranging the paragraphs would clarify the text.
l. 90-91, l. 262-263 - what were the criteria for selecting the nine candidate reference genes for analysis?
Details about calibration curves should be provided.
Results and Discussion
l. 167-169 – general conclusion - The authors should precisely formulate the conclusion whether a combination of reference genes is needed or not.
Minor remarks
Table 1 – the authors should explain abbreviations at the head of the table. In the text the authors should provide information on primers' specificity to potential splice variants and nucleotide polymorphisms.
Figure 3
It is not clear from this figure whether the authors are referring these data to biological replications or biological and technical replications.
Figure 4a, c
The authors should describe Y axis (stability value), and indicate the most and least stable genes.
Figure 4b Y axis – (Gene average of SD analyzed by ΔCT method)? Please indicate the most stable and least stable genes.
